# Aminomethylated Calix[4]resorcinarenes as Modifying Agents for Glycidyl Methacrylate (GMA) Rigid Copolymers Surface

**DOI:** 10.3390/polym11071147

**Published:** 2019-07-04

**Authors:** Betty Astrid Velásquez-Silva, Alver Castillo-Aguirre, Zuly Jenny Rivera-Monroy, Mauricio Maldonado

**Affiliations:** Departamento de Química, Facultad de Ciencias, Universidad Nacional de Colombia-Sede Bogotá, Carrera 30 No. 45-03, 7122 Carrera, Colombia

**Keywords:** resorcinarene, tetrabenzoxazine, tetra-Mannich base, epoxydation

## Abstract

Functionalization of tetrapropylcalix[4]resorcinarene, tetrapentylcalix[4]resorcinarene, tetranonylcalix[4]resorcinarene, and tetra-(4-hydroxyphenyl)calix[4]resorcinarene by means of aminomethylation reactions with the amino acids β-alanine and l-proline in the presence of aqueous formaldehyde was carried out. When β-alanine was used, the reaction products were tetrabenzoxazines. The reaction with tetra-(4-hydroxyphenyl)calix[4]resorcinarene did not proceed under the experimental conditions; therefore, l-proline was used, and the corresponding tetra-Mannich base was regio- and diasteroselectively formed. The products were characterized via FT-IR, ^1^H NMR, ^13^C NMR, and elemental analysis. With these aminomethylated-calix[4]resorcinarenes, the chemical surface modification of the copolymers poly(GMA–*co*–EDMA) and poly(BMA–*co*–EDMA–*co*–MMA) in a basic medium was studied. The results were quite satisfactory, obtaining the corresponding copolymers functionalized by nucleophilic substitution reaction and ring-opening between the carboxyl group of the upper rim of aliphatic calix[4]resorcinarenes and the hydroxyl group of the lower rim in the aromatic calix[4]resorcinarene and the epoxy group of the glycidyl methacrylate residue of each copolymer. The modified copolymers were characterized via FT-IR, scanning electron microscopy imaging, and elemental analysis. Finally, the modified copolymer surfaces exhibited interaction with peptides, showing their potential application in chromatographic separation techniques such as high-performance liquid chromatography.

## 1. Introduction

The development of new polymeric materials for application in chemical separation is currently an active line of investigation in several research groups worldwide [1,2,3,4]. Its importance lies in obtaining new sorbents that significantly improve the analytical characteristics of the determination of analytes of biological interest in complex matrices, such as, for example, greater selectivity and lower LLOQ (lower limit of quantification) [5]. For this purpose, inorganic and organic polymers have been modified with macrocyclic compounds such as cyclodextrins and calixarenes, which can form stable complexes with a variety of analytes [6,7,8]. As a result of these efforts, studies have recently been initiated with the calix[4]resorcinarenes macrocycles as modifiers of polymeric surfaces, mainly because they possess a variety of functional groups that allow selective functionalization, increasing the degree of interaction with the analytes [9]. The versatility of the functionalization of calix[4]resorcinarenes has led to potential applications in other fields, such as chiral nuclear magnetic resonance solvating agents [10,11], chemical receptors for molecules and ions [12,13,14], molecular encapsulation [15,16,17], selective complexation of metal ions and neutral substances [18,19,20], catalysis [21,22], and liquid crystals [23], among others.

Calix[4]resorcinarenes are polyhydroxylated tetrameric platforms that are generally synthesized by the cyclocondensation reaction between resorcinol and an aldehyde, catalyzed by hydrochloric acid [24]. As mentioned above, they have several functional groups, which give them different derivatization alternatives, both on the lower rim and on the upper rim of the macrocyclic structure. Mixed derivatization in the upper rim is also possible, and through the Mannich-type reaction it has been possible to functionalize the aromatic carbons and partially functionalize the hydroxyl groups, obtaining tetrabenzoxazine-type compounds, thus increasing the volume and functionality of the calix[4]resorcinarene cavity in the crown conformation, which has led to their application in analytical separations [9].

In this way, calix[4]resorcinarenes have been successfully applied in the development of new stationary phases for chromatography. Specifically in HPLC, the calix[4]resorcinarenes have been immobilized on polymeric surfaces by post-polymerization modification. Two kinds of functionalization have been used: Firstly, modification of the macrocycle upper rim, by nucleophilic substitution of a hydroxyl group on a highly reactive electrophilic group in the polymer [25]. This methodology is known as “reaction with surface groups”. Using this protocol, silica particles have been chemically modified, obtaining the stationary phases MCR-HPS, BAMCR-HPS [26], and ODS-MCR-HPS [27], which exhibited excellent selectivity and chromatographic behavior in the separation of the position of aromatic isomers and chiral molecules. The second functionalization is the modification of the calix[4]resorcinarene lower rim. Polymers have been modified in two ways. One method is through the lipophilic interaction between long hydrocarbon chains present in the macrocycle and in the polymer. Thus C18 columns have been physically modified with calix[4]resorcinarenes with undecyl groups on the lower rim, showing appropriate selectivity in the separation of nitrogenous bases [9]. The second method is by chemical modification, via a mechanism similar to the modification described for the upper rim. An example is the modification of a RP-C18 column, to which a calix[4]resorcinarene spaced with an undecyl group was covalently linked at the lower rim, observing acceptable selectivity in the separation of *cis* and *trans* isomers of flupentixol, clopenthixol, chlorprothixene, and doxepin [28]. Currently, copolymer materials based on methacrylates have experienced a great upsurge [29], mainly because of the possibility of chemical modification on the epoxy group, when the monomer glycidyl methacrylate is present in the copolymer [30]. In addition, the methacrylate copolymers have the great advantage of being monolithic materials; therefore, they possess high porosity and a large surface area, attributes that are satisfactorily applied in their use as stationary phases.

Continuing with our research on the synthesis and functionalization of calix[4]resorcinarenes [31,32,33,34,35] and on the modification of polymeric surfaces [30], in this article we study the synthesis and use of aminomethylated-calix[4]resorcinarenes in the functionalization of glycidyl methacrylate copolymers and their application in the interaction with peptides.

## 2. Materials and Methods 

### 2.1. General Experimental Information

All the reagents and solvents used in the synthesis experiments were of analytical grade and were used without further purification. A Nicolet^TM^ iS10 FT-IR spectrometer (Thermo Fisher Scientific, Waltham, MA, USA) with a Monolithic Diamond ATR accessory and absorption in cm^−1^ was used for recording IR spectra. Nuclear magnetic resonance (^1^H NMR and ^13^C NMR) spectra were recorded on a BRUKER Avance 400 instrument (400.131 MHz for ^1^H and 100.263 MHz for ^13^C) dimethyl sulfoxide-*d*_6_; the chemical shifts are given in δ units (ppm). RP-HPLC analyses were performed using an Agilent 1200 liquid chromatograph (Agilent, Omaha, NE, USA). The elemental analysis for carbon and hydrogen was carried out using a Flash 2000 elemental analyzer (Thermo Fisher Scientific, Waltham, MA, USA). Scanning electron microscopy (SEM) analysis was done on a VEGA3 SB microscope (TESCAN, Brno-Kohoutovice, Czech Republic).

### 2.2. Synthesis of the Starting Materials

The preparation of calix[4]resorcinarenes has been reported elsewhere [30,34]. Hydrochloric acid (10 mL) was carefully added to a solution of resorcinol (10 mmol) and aldehyde (10 mmol), in a 1:1 ethanol/water mixture (40 mL). The reaction mixture was reacted at reflux with constant stirring for 1–6 h, subsequently cooled in an ice bath, and filtered and washed with water. Finally, the filtrate was dried under a vacuum and was characterized via spectroscopy techniques.

Tetra(propyl)calix[4]resorcinarene (**1**) was obtained as a yellow solid at a yield of 82%, Mp > 250 °C decomposition. IR (KBr/cm^−1^): 3318 (O–H), 2955 (ArC–H), 2929 (aliphatic C–H), 1617 (ArC=C), 1285 (C–O); ^1^H NMR, DMSO-*d*_6_, δ (ppm): 0.90 (t, 12H, CH_3_), 1.20 (m, 8H, CH_2_), 2.08 (q, 8H, CH_2_), 4.23 (t, 4H, CH), 6.15 (s, 4H, *ortho* to OH), 7.24 (s, 4H, *meta* to OH), 8.93 (s, 8H, OH); ^13^C NMR, δ (ppm): 16.1, 25.3, 32.7, 49.2, 112.1, 127.2, 129.0, 151.1.

Tetra(pentyl)calix[4]resorcinarene (**2**) was obtained as a yellow solid at a yield of 83%, Mp > 250 °C decomposition. IR (KBr/cm^−1^): 3416 (O–H), 2950 (ArC–H), 2928–2858 (aliphatic C–H), 1620 (ArC=C), 1292 (C–O); ^1^H NMR, DMSO-*d*_6_, δ (ppm): 0.85 (t, 12H, CH_3_), 1.19 (m, 8H, CH_2_), 1.28 (m, 16H, CH_2_), 2.03 (m, 8H, CH_2_), 4.24 (t, 4H, CH), 6.17 (s, 4H, *ortho* to OH), 7.16 (s, 4H, *meta* to OH), 8.90 (s, 8H, OH); ^13^C NMR, δ (ppm): 14.1, 22.4, 27.6, 31.6, 33.1, 34.1, 102.5, 123.2, 124.9, 151.8.

Tetra(nonyl)calix[4]resorcinarene (**3**) was obtained as a yellow solid at a yield of 87%, Mp > 250 °C decomposition. IR (KBr/cm^−1^): 3258 (O–H), 2925 (ArC–H), 2853 (aliphatic C–H), 1620 (ArC=C), 1294 (C–O); ^1^H NMR, DMSO-*d*_6_, δ (ppm): 0.83 (t, 12H, CH_3_), 1.20 (br. s., 56H, CH_2_), 1.96 (br. s., 8H, CH_2_), 4.21 (t, 4H, CH), 6.15 (s, 4H, *ortho* to OH), 7.07 (s, 4H, *meta* to OH), 8.88 (s, 8H, OH); ^13^C NMR, δ (ppm): 13.8, 22.1, 27.8, 28.8, 28.9, 29.2, 29.2, 29.3, 31.4, 34.2, 102.4, 123.0, 124.3, 151.7.

Tetra(4-hydroxyphenyl)calix[4]resorcinarene (**4**) was obtained as a pink solid at a yield of 54%, Mp > 250 °C decomposition. IR (KBr/cm^−1^): 3384 (O–H), 1249 (C–O); ^1^HNMR, DMSO-*d*_6_, δ (ppm): 5.53 (s, 4H, ArCH), 6.08 (s, 4H, ArH, *ortho* to OH), 6.47 (d, 8H, J = 8 Hz, ArH), 6.50 (s, 4H, ArH, *meta* to OH), 6.63 (d, 8H, J = 8 Hz, ArH), 8.43 (s, 8OH, ArOH), 8.83 (4OH, ArOH); ^13^C NMR, δ (ppm): 40.6, 102.9, 114.0, 121.0, 129.6, 136.0, 152.2, 152.3, 154.5.

### 2.3. Reaction of Calix[4]resorcinarenes **1**–**3** with β-alanine

The preparation of benzoxazines from calix[4]resorcinarenes **1**–**3**, formaldehyde, and β-alanine was performed using the following methodology: To a solution of calix[4]resorcinarene (**1**, **2** or **3**) (1 mmol) in 44 mL of a mixture C_6_H_6_:EtOH (1:1), a solution of β-alanine (4 mmol) and formaldehyde (10 mmol) was added. The reaction mixture was reacted at reflux with constant stirring for 12–29 h. The product was filtered, washed with ethanol, and dried under a vacuum. The products were characterized via spectroscopy techniques. 

2,12,22,32-tetrapropyl-4,14,24,34-tetrahidroxy-7,17,27,37-tetra-(2-carboxyethyl)-7,17,27,37-tetra-azanonacycle[31.3(7).1.1^3,11^.1^13,21^.1^23,31^1(41).3.5(10)-11(44).13.1^23,31^.0^5,10^.0^15,20^.0^25,30^.0^35,40^]tetraconta[15(20), 21(43),23,25(30),31(42),33,35(41)]dodecaene (**5**):

An purple solid at yield of 88%. Mp > 250 °C decomposition. IR (KBr/cm^−1^): 3374 (O–H y COOH), 2955 (Ar. C–H), 2869 (aliphatic C–H), 1714–1603 (C=O), 1214 (C–N), 1115 (C–O); ^1^H NMR, DMSO-*d*_6_, δ (ppm): 0.93 (br. s., 12H, CH_3_), 1.25 (br. s., 8H, CH_2_), 2.20 (br. s., 8H, CH_2_), 2.43 (br. s., 8H, NCH_2_–**CH_2_**–COOH), 2.90 (br. s., 8H, N–**CH_2_**–CH_2_COOH), 3.97 (br. s., 8H, ArCH_2_N), 4.27 (br. s., 4H, CH), 6.20 (br. s., 12H, NCH_2_O and OH), 7.26 (br. s., 4H, *meta* to OH); ^13^C NMR, δ (ppm): 13.9, 20.8, 31.7, 33.6, 35.4, 42.9, 56.0, 85.0, 107.6, 123.6, 124.5, 152.5, 152.6, 172.9, 174.1. Anal. calcd. for (molecular formula, C_60_H_76_N_4_O_16_): C = 64.97, H = 6.91 and N = 5.05; found: C = 60.90, H = 6.90 and N = 4.41 (because of its host nature, the compound occluding solvent molecules observed by NMR).

2,12,22,32-tetrapentyl-4,14,24,34-tetrahidroxy-7,17,27,37-tetra-(2-carboxyethyl)-7,17,27,37-tetra-azanonacycle[31.3(7).1.1^3,11^.1^13,21^.1^23,31^1(41).3.5(10)-11(44).13.1^23,31^.0^5,10^.0^15,20^.0^25,30^.0^35,40^]tetraconta[15(20), 21(43),23,25(30),31(42),33,35(41)]dodecaene (**6**):

An purple solid at a yield of 89%. Mp > 250 °C decomposition. IR (KBr/cm^−1^): 3411 (O–H and COOH), 2926 (ArC–H), 2855 (aliphatic C–H), 1722–1607 (C=O), 1211 (C–N), 1109 (C–O); ^1^H NMR, DMSO-*d*_6_, δ (ppm): 0.85 (br. s., 12H, CH_3_), 1.27 (br. s., 16H, CH_2_), 2.18 (br. s., 8H, CH_2_), 2.40 (br. s., 8H, CH_2_), 2.86 (br. s., NCH_2_–**CH_2_**–COOH), 2.94 (br. s., N–**CH_2_**–CH_2_COOH), 3.94 (br. s., 8H, ArCH_2_N), 4.19 (br. s., 4H, CH), 4.67 (br. s., NCH_2_O), 4.85 (br. s., OH), 7.20 (br. s., 4H, *meta* to OH). Anal. calcd. for (molecular formula, C_68_H_92_N_4_O_16_): C = 66.86, H = 7.59 and N = 4.59; found: C = 63.35, H = 7.51 and N = 4.05 (because of its host nature, the compound occluding solvent molecules observed by NMR).

2,12,22,32-tetranonyl-4,14,24,34-tetrahidroxy-7,17,27,37-tetra-(2-carboxyethyl)-7,17,27, 37-tetra-azanonacycle[31.3(7).1.1^3,11^.1^13,21^.1^23,31^1(41).3.5(10)-11(44).13.1^23,31^.0^5,10^.0^15,20^.0^25,30^.0^35,40^]tetraconta[15(20), 21(43),23,25(30),31(42),33,35(41)]dodecaene (**7**):

An purple solid at a yield of 77%. Mp > 250 °C decomposition. IR (KBr/cm^−1^): 3279 (O–H and COOH), 2922 (ArC–H), 2889 (aliphatic C–H), 1714–1620 (C=O), 1212 (C–N), 1103 (C–O); ^1^H NMR, DMSO-*d*_6_, δ (ppm): 0.86 (br. s., 12H, CH_3_), 1.24 (br. s., 64H, CH_2_), 2.18 (br. s., 8H, NCH_2_–**CH_2_**–COOH), 2.33 (br. s., 8H, NCH_2_–**CH_2_**–COOH), 2.84 (br. s., ArCH_2_N), 3.87 (br. s., CH), 4.04 (br. s., OH), 4.16 (br. s., NCH_2_O), 7.05 (br. s., 2H, *meta* to OH), 7.13 (br. s., 2H, *meta* to OH). Anal. calcd. for (molecular formula, C_84_H_124_N_4_O_16_): C = 69.78, H = 8.64 and N = 3.87; found: C = 68.17, H = 8.80 and N = 4.05 (because of its host nature, the compound occluding solvent molecules observed by NMR).

### 2.4. Reaction of Calix[4]resorcinarene **4** with l-proline

EtOH (8 mL) and 37% aqueous formaldehyde (2 mmol, 190 μL) were added to a solution of (**4**) (0.2 mmol) in DMF (5 mL) in a 150% stoichiometric excess under constant stirring [36,37]. Subsequently, a solution of l-proline (1 mmol) containing NaOH 1 M (43 μL) in water (700 μL) was added. The reaction was left at room temperature for 6 h under magnetic stirring, whereupon an amorphous solid was formed, which was filtered under a vacuum and washed with EtOH (5 mL). The precipitate was dried at room temperature for 24 h in the absence of light. The product was characterized via FT-IR, ^1^H NMR, ^13^C NMR, MS, elemental analysis, and HPLC. The following was obtained:

2,8,14,20-Tetra-(4-hydroxyphenyl)-5,11,17,23-tetrakis(*N*–(l-proline)methyl)pentacyclo[19.3.1. 1^3;7^.1^9;13^.1^15;19^]octacosa-1(25),3,5,7(28),9,11,13(27),15,17,19(26),21,23-dodecaen-4,6,10,12,16,18,22,24-octol (**8**): 

An ocher yellow solid at a yield of 87%. Mp > 250 °C (desc.). IR (KBr/cm^−1^): 3413 (O–H), 1678 (C=O). ^1^H NMR, DMSO-*d_6_*, δ (ppm): 1.62–1.87 (m, 16H, proline NC–(CH_2_)_2_), 2.11–2.20 (m, 8H, proline NCH_2_), 3.88 (m, 4H, proline CH), 3.97 (m, 8H, NCH_2_Ar), 5.54–5.76 (m, 4H, ArCH), 6.20–6.55 (m, 20H, ArH). ^13^C NMR, δ (ppm): 22.8 and 23.4, 28.3 and 28.7, 41.9 and 42.2 (proline N(CH_2_)_3_), 49.7 and 49.9 (ArCH), 52.4 and 53.1 (ArCH_2_N), 67.1 and 67.4 (proline CH), 108.5 and 114.2 (resorcinol C–2), 121.8 and 122.9 (resorcinol C–5), 122.9 and 123.1 (hydroxyphenyl C–3), 129.6 and 129.7 (hydroxyphenyl C–2), 132.6 and 133.2 (hydroxyphenyl C–4), 150.9 and 151.9 (resorcinol C–4), 152.0 and 152.2 (hydroxyphenyl C–1), 154.8 and 154.9 (resorcinol C–1), 172.3 and 172.9 (C=O). Anal. calcd. for (molecular formula, C_76_H_76_N_4_O_20_): C = 61.95, H = 6.02 and N = 3.80; found: C = 61.30, H = 6.30 and N = 3.80 (because of its host nature, the compound occluding solvent molecules observed by NMR).

### 2.5. Preparation of Copolymers

#### 2.5.1. Poly(GMA–*co*–EDMA)

The preparation of copolymer has been reported elsewhere [38]. The monomer inhibitor (hydroquinone monomethyl ether, MEHQ) was removed before performing the polymerization reaction, using a silica gel column. A mixture of GMA (0.34 mL), EDMA (0.23 mL), 1,1-azobis(cyclohexanecarbonitrile) (6.3 mg), cyclohexanol (1.22 mL), and dodecanol (0.27 mL) was prepared, purged with N_2_ for 5 min, and then incubated at 57 °C for 24 h. Next, the resulting polymer was cooled to room temperature, porogens were removed by washing with absolute ethanol, and the polymer was dried under reduced pressure. The copolymers were characterized via IR (ATR), elemental analysis, and SEM.

#### 2.5.2. Poly(BMA–*co*–EDMA–*co*–GMA)

The preparation of a copolymer was performed in the same way as the previous procedure, and the amounts of the monomers were determined according to previous studies [39]. BMA (0.15 mL), GMA (0.45 mL), EDMA (0.40 mL), 1,1-azobis(cyclohexanecarbonitrile) (11 mg), cyclohexanol (1.50 mL), and dodecanol (0.15 mL).

### 2.6. Surface Modification of Copolymers with Aminomethylated-Calix[4]resorcinarenes **5**–**7**

For the surface modification of copolymers poly(BMA–*co*–EDMA–*co*–GMA) and poly(GMA–*co*–EDMA) with benzoxazines calix[4]resorcinarene **5**–**7**, the methodology was adapted from the procedure published by Zargin et al. [40]. To a solution of benzoxazine calix[4]resorcinarene (1 mmol) in DMF, a solution of tetrabuthylammonium chloride (1 mmol) in DMF was added dropwise, and then this mixture was added dropwise to a copolymer suspension in DMF. Afterward, the reaction mixture was purged with N_2_, sealed, and reacted for 24 h with constant stirring. Then the reaction mixture was filtered, washed with DMF and absolute ethanol, and dried under reduced pressure. Six modified copolymers with benzoxazines calix[4]resorcinarene were obtained and characterized via IR (ATR), elemental analysis, and SEM. 

### 2.7. Surface Modification of Copolymers with Aminomethylated-Calix[4]resorcinarene **8**

The functionalization process was adapted from the methodology previously proposed by us [30]. A mixture of (**1**) (0.2 mmol) in DMF/DMSO and sodium hydroxide (0.2 mmol) was added to triturated copolymer poly(GMA–*co*–EDMA–*co*–MMA) (500 mg). The rest of the procedure followed the same as the previous methodology.

### 2.8. Interaction of Peptides with Copolymers

Calibration curves were performed for the peptides via RP-HPLC, using: (i) a Chromolith® C–18 column (4.6 × 50 mm) as the stationary phase; (ii) an elution gradient of 5% to 50% of solvent B was performed for 8 min—specifically, solvent A corresponded to 0.05% TFA in H_2_O and solvent B was 0.05% TFA in ACN; (iii) a flow of 2 mL/min; (iv) 210 nm for detection; and (v) an injection volume of 10 μL.

To determine the interaction between peptides and copolymers, 1.0 mg of the tested copolymer was added to the peptide solution (1 mg/mL) and gently stirred for 1 min, and then it was centrifuged, and an aliquot was analyzed via HPLC in order to calculate the peptide concentration without interacting. These analyses were performed in an acidic and a neutral medium. The percentage of interaction with the modified copolymer was calculated using the substitution of the resorcinarene (Table 1) as the maximum value and considering that the relation of peptide and resorcinarene motif was 1:1.

## 3. Results

### 3.1. Synthesis of Calix[4]resorcinarenes and Aminomethylated Derivates

As mentioned above, the experimental process was carried out in several stages. The first step involved the synthesis of calix[4]resorcinarenes, and for the present study we chose four calix[4]resorcinarenes (**1**–**4**) to be modified by aminomethylation reaction and subsequently anchored to the polymeric surface. The synthesis of calix[4]resorcinarenes **1**–**4** was done through the acid-catalyzed cyclocondensation of resorcinol with an aldehyde in a 50:50 mixture of ethyl alcohol and water at 75 °C (Scheme 1). The products were purified by means of recrystallization, and the derivatives were characterized using spectral techniques, including FT-IR, ^1^H NMR, ^13^C NMR, and mass spectrometry. These compounds have been previously synthesized, and the spectroscopic data agreed with those reported by us [30,31,33,34].

The next step involved reaction with amino acids in the presence of formaldehyde via base-catalyzed cyclocondensation, as described in the literature [36]. Briefly, first the reaction of β-alanine was carried out with the calix[4]resorcinarenes **1**–**3** (see Scheme 2). The products obtained in each reaction were easily purified by recrystallization and were characterized using spectral techniques including elemental analysis, MS-spectrometry, FT-IR, ^1^H NMR, and ^13^C NMR experiments (see the experimental section). In this way, the aminomethylation product of **1** and **3** showed that the FT-IR was in agreement with the organic functionalities present, the principal features observed being the hydroxyl group stretches at 3347 cm^−1^ (O–H) and 1468 cm^−1^ (C−O), whereas the C–N of the amine group can be seen at 1286 cm^−1^. The bands of the alkyl substituent and the aromatic ring can also be seen at 2923 and 3027 cm^−1^, respectively. The ^1^H NMR spectrum of the products **5**–**7** displayed characteristic signals of benzoxazine ring in a range of 3.87–3.97 ppm (Ar–CH_2_–N) and 4.16–4.67 ppm (N–CH_2_–O). Likewise, characteristic signals for β-alanine moiety were observed in a range of 2.18–2.86 ppm, and signals of alkyl substituent were observed in the range of 1.28–2.17 ppm (see experimental section for the complete assignment of each product). Methine bridges in the three products were also observed (4.16–4.27 ppm). In the aromatic region, a singlet signal for *meta* protons of the three calix[4]resorcinarenes′ moiety was detected in the range of 7.13–7.26 ppm. The signal for the hydroxyl groups was at 7.66 ppm, but it only integrated for four protons. So, the decrease of the integral of the hydroxyl signals is indicative of the formation of a benzoxazine ring. The ^13^C NMR spectra of **5**–**7** displayed the number of signals expected for each compound, and they were assigned according to their displacements and based on previous reports (see experimental section) [31,34,41,42].

Aminomethylation of **4** was done with proline by direct reaction with formaldehyde solution (Scheme 3). In this way, compound **8** was obtained as a white solid, and the molecular weight was determined via MALDI-TOF-MS (*m/z* 1525.20 corresponding to the specie [M + Na]^+^). The FT-IR spectrum of **8** showed amine group (044 cm^−1^ and 773 cm^−1^), aromatic ring (1609 cm^−1^), alkyl chain (2929 cm^−1^), and hydroxyl group (3421 cm^−1^) absorptions. The ^1^H NMR spectrum displayed characteristic signals for the calix[4]resorcinarene system: a methine bridge fragment between the aromatic rings (3.97 ppm) and the aromatic hydrogen of a pentasubstituted resorcinol unit (6.50 ppm). The strong cyclic hydrogen bonding in the product is the response of the signals for diastereotopic protons Ar–CH_2_–N at 5.54 and 5.76 ppm, the two signals with doublet multiplicity and a coupling constant of *J* = 16 Hz. The signals for proline moiety were displayed at 1.62, 2.11, and 2.20 ppm. The carbon signals in ^13^C NMR were assigned through 1D experiments and based on previously published articles [31,34,41,42] (see experimental section).

### 3.2. Copolymerization and Surface Modification

Once the calix[4]resorcinarenes **5**–**8** were obtained, the next step involved the reaction of these macrocycles with the copolymers′ poly(BMA–*co*–EDMA–*co*–GMA) and poly(EDMA–*co*–GMA) surface. Thus, the copolymers were obtained according to previous reports [30]. Characterization of the copolymers was done via ATR-FTIR and scanning microscopy, and the spectra show a characteristic signal correspond ding to a C=O bond at 1714 cm^−1^. In addition, other characteristic peaks were found at 2930 and 104 cm^−1^ due to the C–H stretching vibration, 1449 cm^−1^ due to the C–H bending vibration, 1151 cm^−1^ due to the C–O stretching vibration of the glycidyl group, and 902 cm^−1^ due to characteristic epoxy ring absorption. The results showed that there was an absence of the C=C peak at 1600–1675 cm^−1^, which suggested that the polymerization method was effective. The 2,3-epoxypropyl methacrylate-copolymer morphology was analyzed via scanning electron microscopy imaging. It was observed that the morphology of the 2,3-epoxypropyl methacrylate-copolymer is quite uniform, with continuous pores, this kind of structure is suitable for mass transfer efficiency for the reaction between the oxirane side chain of copolymers and the calix[4]resorcinarenes **5**–**8**.

The presence of an oxirane ring in 2,3-epoxypropyl methacrylate copolymers allows further chemical modifications with various reactive groups. In recent years, research has been carried out on the attachment of a hydroxyl group to copolymers of 2,3-epoxypropyl methacrylate and with the carboxylate group [40]. For the present investigation, we performed the copolymer grafting by two routes: first, fixation of each calix[4]resorcinarene molecule (**5**–**7**) by its direct reaction with the copolymer’s glycidyl group in the presence of tetrabuthylammonium chloride in DMF via esterification with the carboxylic group in the upper rim in the calix[4]resorcinarenes (**5**–**7**), and second by fixation of calix[4]resorcinarene **8** by its direct reaction with the copolymer's glycidyl group in basic media via attachment of a hydroxyl group of **8** in the lower rim to the 2,3-epoxypropyl group of copolymers (Scheme 4).

For the first route, the products obtained from the reaction between copolymers and calix[4]resorcinarenes **5**–**8** exhibit ATR-FTIR spectra showing the presence of broad and intense absorption from 3220 to 3432 cm^−1^ corresponding to the O–H groups of calix[4]resorcinarenes rings, the presence of intense peaks around 1651 cm^−1^ and 1591 cm^−1^ corresponding to C=C of aromatic ring, and a peak between 1144 to 1170 cm^−1^, attributed to C–O of phenolic residue. The absorption at 906 cm^−1^ exhibited a reduction in its intensity for the modified copolymer spectrum compared to the unmodified one, suggesting that epoxy ring was opened during the reaction, and this result indicated that calix[4]resorcinarenes **5**–**8** were chemically bound to copolymers. As an example, in Figure 1, it is shown the IR spectra of the starting copolymer and the monolith modified with **5**. Using the second route, it was observed that the binding of calix[4]resorcinarene **8** occurs successfully on the copolymer poly(BMA–*co*–EDMA–*co*–MMA), while with the copolymer poly(GMA–*co*–EDMA) the reaction does not occur. These results suggest that in the case of **8**, the polymer must have a spacer (MMA), which considerably facilitates the process of grafting. According to the results obtained with these processes, it was indeed possible to obtain surfaces modified on the upper rim of resorcinarenes **5**–**7** and modification of a surface on the lower rim with resorcinarene **8** (Scheme 4), which was efficiently obtained. The characterization data obtained with all the modified copolymers is shown in Table 1. 

The 2,3-epoxypropyl methacrylate-copolymers’ morphology, before and after modification with calix[4]resorcinarenes **5**–**8**, was analyzed via scanning electron microscopy imaging (Figure 2). It was observed that the morphology is quite uniform, with continuous pores; modification did not change the morphology of the copolymers. The modified copolymers were exhaustively washed, and the degree of aminomethylated-calix[4]resorcinarenes (**5**–**8**) incorporation was determined from the information of the nitrogen percentage in the elemental analysis, i.e., the original copolymer did not contain nitrogen so the source of it is the aminomethylated-calix[4]resorcinarene attached to the surface, in all cases those molecules contain four nitrogen atoms and this information was used to calculate the degree of substitution at the copolymer (µmol of resorcinarene/g of copolymer), as it is shown in Table 1. These results show that the copolymers poly(BMA–*co*–EDMA–*co*–GMA) have the highest degree of substitution (Table 1, VI-IX), which allows suggesting that the presence of BMA acts as a spacer, allowing better surface modification using the macrocycles.

### 3.3. Interaction of the Functionalized Surfaces with Peptides 

The interaction of the modified material **6**-poly(BMA–*co*–EDMA–*co*–GMA) with two peptides was tested, specifically using AcEYSFEFSY and KAEAEAKA sequences. The experiments were carried out under two conditions of pH, and the determination of the interaction was performed using the obtained copolymers in batch, with the measurement performed indirectly.

Thus, a solution of each peptide (1.0 mg/mL) was prepared, and a for each molecule a calibration curve was done. The solution was treated with 1.0 mg of the copolymer and then was centrifugated and analyzed via RP-HPLC. The peptide area was used for calculating the concentration of peptide remaining in the solution, and the retained quantity was compared with the maximum quantity that could be absorbed. As is shown in Table 2, peptide AcEYSFEFSY exhibits greater interaction with the modified copolymer **6**-poly(BMA–*co*–EDMA–*co*–GMA) than peptide KAEAEAKA. Its interaction was greater at neutral pH. The interaction of this peptide with the modified copolymer could be due to the presence of aromatic rings from the Tyr residue with the resorcinarene motifs present at the surface, via π–π interaction. On the other hand, the copolymer without modification exhibited a weak interaction with both peptides at acidic and neutral conditions.

## 4. Conclusions

Aminomethylation reactions with amino-compounds and aqueous formaldehyde were efficiently carried out with tetraproylcalix[4]resorcinarene (**1**), tetrapentylcalix[4]resorcinarene (**2**), tetranonylcalix[4]resorcinarene (**3**), and tetra(*p*-hydroxyphenyl)calix[4]resorcinarene (**4**) with percentage yields between 54% and 87%. The reaction of the calix[4]resorcinarenes **1**–**3** with β-alanine generated tetrabenzoxazines **5**–**7**, whereas the reaction of the calix[4]resorcinarene **4** with l-proline generated tetraMannich base **8**. 

To evaluate the efficiency of the surface modification with calix[4]resorcinarenes **5**–**8**, a copolymer with two components, 2,3-epoxypropyl methacrylate and ethylene glycol dimethacrylate (GMA–*co*–EDMA), was produced and characterized via scanning electron microscopy (SEM) and ATR-FTIR spectroscopy. The copolymer was treated under N_2_ atmosphere with aminomethylated- calix[4]resorcinarenes **5**–**7** in DMF and tetrabuthylammonium chloride**.** ATR-FTIR spectroscopy results showed that macrocycles were covalently bonded to the copolymer, indicating that the reaction is selective towards the carboxylic groups of the upper rim of the resorcinarenes. Surface modification of copolymer poly(GMA–*co*–EDMA) with calix[4]resorcinarene **8** was done between the epoxide ring and hydroxyl groups of the lower rim of **8** under basic conditions.

As a preliminary evaluation of the potential use of modified copolymers as stationary phases, the capacity of the monolith, functionalized with the resorcinarene motif, to trap peptides of different nature at two pH values (acidic and neutral) was tested, and it was found that the peptide with aromatic rings exhibits a greater interaction with this material.

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
