# Peer review of "Aminomethylated Calix[4]resorcinarenes as Modifying Agents for Glycidyl Methacrylate (GMA) Rigid Copolymers Surface"

_polymers, 2019, doi:10.3390/polym11071147_

Reviewer 1 Report

The authors present a manuscript detailing the surface modification methacrylate copolymers with calix[4]resorcinarene additives. The authors make this modification with aminomethylation reaction with L-proline and the resulting materials was tested.The manuscript is well written and readily understood. The language used is very good. Overall the presented research could be valuable to other researchers in the field of surface modification of polymers.  The researchers presented a good introduction of the problem and previous work by other researchers. The researchers were very careful in describing the experimental part. Therefore, these experiments could be replicated by other researchers familiar with the area. The conclusions in this manuscript are supported by the experimental results. The researchers tested the materials by FTIR, NMR, SEM, and elemental analysis. Thus I support the publication of this manuscript.

*remove text of supplemental materials.

Bogotá D.C. 28th June 2019

Referee: 1

The authors present a manuscript detailing the surface modification methacrylate copolymers with calix[4]resorcinarene additives. The authors make this modification with aminomethylation reaction with L-proline and the resulting materials was tested.The manuscript is well written and readily understood. The language used is very good. Overall the presented research could be valuable to other researchers in the field of surface modification of polymers.  The researchers presented a good introduction of the problem and previous work by other researchers. The researchers were very careful in describing the experimental part. Therefore, these experiments could be replicated by other researchers familiar with the area. The conclusions in this manuscript are supported by the experimental results. The researchers tested the materials by FTIR, NMR, SEM, and elemental analysis. Thus I support the publication of this manuscript.

Answer: We appreciate the reviewer comments, and we thank the manuscript review

Reviewer 2 Report

The paper reports on the functionalization of GMA copolymers with calix[4]resorcinarene units targeting solid support for chromatographic separation techniques.

The work is well performed in terms of calix[4]resorcinarene macrocycles synthesis but their attachment to the copolymeric materials should be strengthen. More specifically FTIR as SEM do not give any quantitative indication for the extent of the modification. Additional experiments using thermogravimetric analysis, or even weight increase data after modification reactions could help on the better characterization.

Additionally the very preliminary results, on the interaction with peptides, which are provided should also be enriched. A minor point is that all schemes are appeared in the text as Diagrams.

Overall, the manuscript needs major revision before acceptance.

Bogotá D.C. 28th June 2019

Referee: 2

The paper reports on the functionalization of GMA copolymers with calix[4]resorcinarene units targeting solid support for chromatographic separation techniques.The work is well performed in terms of calix[4]resorcinarene macrocycles synthesis but their attachment to the copolymeric materials should be strengthen.More specifically FTIR as SEM do not give any quantitative indication for the extent of the modification. Additional experiments using thermogravimetric analysis, or even weight increase data after modification reactions could help on the better characterization. Additionally, the very preliminary results, on the interaction with peptides, which are provided should also be enriched. A minor point is that all schemes are appeared in the text as Diagrams.

Answer: We agree with the Reviewer comments.

1. We include a new paragraph discussing  the "quantitative indication for the extent of the modification" Please see line 325-333:

The modified copolymers were exhaustively washed, and the degree of aminomethylated-calix[4]resorcinarenes 5-8 incorporation was determined from the information of the nitrogen percentage in the elemental analysis, i.e. the original copolymer did not contain nitrogen so the source of it is the aminomethylated-calix[4]resorcinarene attached to the surface, in all cases those molecules contain four nitrogen atoms and this information was used to calculate the degree of substitution at the copolymer (µmol of resorcinarene/g of copolymer), as it is  shown in Table 1. These results show that the copolymers poly(BMA-co-EDMA-co-GMA) have the highest degree of substitution (Table 1, VI-IX), which allows suggesting that the presence of the mesomeric unit of BMA acts as a spacer, allowing better surface modification using the macrocycles.

2. All the words Diagram were changed by Scheme through the text

Round  2

Reviewer 2 Report

After the authors clarifications the manuscript can be accepted in the present form.